# Transcriptome Sequencing-Based Screening of Key Melatonin-Related Genes in Ischemic Stroke

**DOI:** 10.3390/ijms252111620

**Published:** 2024-10-29

**Authors:** Tianzhi Li, Hongyan Li, Sijie Zhang, Yihan Wang, Jinshan He, Jingsong Kang

**Affiliations:** Key Laboratory of Pathobiology, Department of Pathophysiology, Ministry of Education, College of Basical Medical Sciences, Jilin University, 126 Xinmin Street, Changchun 130012, China; tzli23@mails.jlu.edu.cn (T.L.); hongyan@jlu.edu.cn (H.L.); sijie23@jlu.edu.cn (S.Z.); yihanw22@jlu.edu.cn (Y.W.); hejs24@jlu.edu.cn (J.H.)

**Keywords:** ischemic stroke, melatonin, biomarker, single cell, transcriptome

## Abstract

Ischemic stroke (IS) is a complex syndrome of neurological deficits due to stenosis or occlusion of the carotid and vertebral arteries for which there is still no effective treatment. Melatonin, a hormone secreted by the pineal gland, has multiple biological effects, such as antioxidant and anti-inflammatory properties, circadian rhythm regulation, and tissue regeneration, demonstrating potential applications in the treatment of IS. The aim of this study was to investigate key melatonin-regulated genes associated with IS using transcriptome sequencing and bioinformatics analyses and to explore their potential mechanisms of action in the disease process. We obtained gene expression data related to ischemic stroke (IS) from the Gene Expression Omnibus (GEO) database and identified candidate genes using machine learning algorithms. We then assessed the predictive power of these genes using PPI network analysis and diagnostic models. Finally, a series of enrichment analyses identified four key genes: ADM, PTGS2, MMP9, and *VCAN*. In addition, we determined the mRNA levels of these four key genes in an IS rat model using qPCR and found that all of these genes were significantly upregulated in the IS model compared to the control group, which is consistent with the results of previous analyses. Meanwhile, these genes have biological functions such as regulating vascular tone, participating in the inflammatory response, influencing tissue remodeling, and regulating cell adhesion and proliferation, playing key roles in the pathogenesis of IS. Therefore, we suggest that these four key genes may serve as prospective biomarkers for IS and help predict the risk of developing IS. In conclusion, this study elucidates for the first time the potential role of melatonin in the pathogenesis of IS and lays the foundation for in-depth studies on the functions of these key genes in the pathophysiology of IS and their potential applications in clinical diagnosis and treatment.

## 1. Introduction

Stroke is an acute cerebrovascular disease characterized by the sudden rupture or blockage of a cerebral blood vessel, resulting in tissue damage within the brain [1,2]. Approximately 795,000 strokes occur annually in the United States, of which approximately 690,000 (87%) are classified as ischemic strokes, and 185,000 are classified as recurrent strokes [3]. Cerebral ischemia induces a series of biochemical and cellular responses that generate excessive reactive oxygen species (ROS) [4]. When endogenous antioxidant capacity is insufficient to maintain redox homeostasis, oxidative stress occurs, causing cytotoxicity with deleterious consequences for brain tissue structure and function [5,6]. At the same time, ischemic injury leads to the release of a variety of inflammatory mediators such as cytokines and chemokines [7]. Inflammatory cells such as macrophages and T cells are activated to produce more inflammatory mediators, and the inflammatory response further exacerbates neuronal damage. Moreover, mitochondrial dysfunction and activation of proteases due to ischemia lead to apoptosis of neuronal cells, exacerbating brain damage [8].

Currently, there are two main treatments for ischemic stroke: thrombolysis and thrombectomy. The mainstay of thrombolysis is intravenous recombinant tissue plasminogen activator (rt-PA), which must be administered within 4.5 h to prevent hemorrhagic transformation [9]. Although several treatment options are available for stroke patients, the efficacy of these treatments is limited by the physiological processes that occur after stroke and by patient factors such as age, gender, and pathological diversity [10,11]. Ischemic stroke is a polygenic disease with a significant genetic component [12], while multiple cumulative risk factors and causes contribute to the heterogeneity of stroke [13]. As stroke is one of the most clinically disabling and fatal diseases, its diagnosis and treatment currently rely heavily on clinical trials. Response rates are inadequate and the recurrence of symptoms is more common in patients who discontinue treatment. Accurate diagnosis of stroke and early prevention are essential to reduce suffering and improve prognosis. Therefore, new features or biomarkers need to be identified to improve clinical decision making in stroke management.

Melatonin is a hormone secreted by the pineal gland that has a variety of properties including antioxidant and anti-inflammatory effects, circadian regulation, and tissue regeneration. Many experiments have confirmed that melatonin exerts significant neuroprotective effects at various stages of stroke by downregulating reactive oxygen species levels through scavenging free radicals, upregulating the expression and activity of endogenous antioxidants, and attenuating inflammatory responses [14]. Pei et al. reported that prophylactic treatment with melatonin reduced infarct volume in a rat model of middle cerebral artery occlusion (MCAO) [15]. Feng et al. showed that pre-ischemic melatonin treatment could exert neuroprotective effects through inhibition of ER stress-related autophagy pathways and that this neuroprotection included a reduction in infarct volume and cerebral water content, as well as an improvement in neurological scores over the two-week post-ischemic period and in survival [16]. In a study of long-term melatonin therapy, Chen et al. used gerbils (members of the mouse family lacking posterior communicating arteries) in a transient global cerebral ischemia (tGCI) model, administered melatonin for 25 consecutive days, and showed that long-term melatonin therapy improved cognitive deficits after tGCI [17]. Meanwhile, another study reported that melatonin administration modulated microglia/macrophage polarization toward an anti-inflammatory phenotype via the STAT3 pathway, thereby attenuating the pro-inflammatory response and improving functional outcomes in a mouse model of distal MCAO (dMCAO) [18]. However, little is known about validated diagnostic biomarkers of melatonin in the context of stroke, and further clarification is required.

This study aimed to investigate altered gene expression in the pathophysiology of ischemic stroke (IS) and develop novel potential diagnostic biomarkers. Through analysis of two GEO datasets, we identified 74 differentially expressed genes from IS samples. These differentially expressed genes were intersected with melatonin-related genes, resulting in the identification of 35 hub genes. Using support vector machine-recursive feature elimination (SVM-RFE), random forest (RF), and least absolute shrinkage and selection operator (LASSO) algorithms, we classified and screened these hub genes, ultimately identifying four key genes: ADM, MMP9, VCAN, and PTGS2. To validate our model’s accuracy and generalizability, we employed an external validation dataset and performed receiver operating characteristic (ROC) curve analysis. Subsequently, we analyzed the intracellular expression distribution of these key genes using single-cell sequencing data. In addition, research indicates that melatonin regulates the expression of adrenomedullin (ADM), which is crucial for vasodilation and neuroprotection [19]. It also influences vascular proteoglycan (VCAN), which is involved in extracellular matrix remodeling and neural repair [20], and regulates matrix metalloproteinase 9 (MMP9), which is essential for blood–brain barrier integrity and neuroinflammation [21]. Furthermore, melatonin inhibits cyclooxygenase-2 (PTGS2), reducing inflammatory responses and oxidative stress [22]. The interactions between these genes and melatonin provide new perspectives for understanding the protective mechanisms of melatonin in ischemic stroke. At the same time, the differential genes were submitted to the CMAP database to predict potential small molecule regulators. Finally, we validated the expression differences of the key genes in normal and IS samples using qPCR to explore their potential as diagnostic and therapeutic targets for IS.

## 2. Results

### 2.1. Identification of Differential Genes

In this study, two microarray datasets, GSE16561 and GSE22255, were merged and analyzed differentially after removing batch effects. A total of 74 differentially expressed genes (DEGs) were identified in the integrated expression matrix by selecting *p* < 0.05 and |log fold change (FC)| > 0.5 as thresholds (see Appendix A). Volcano and heat maps were drawn for these differential genes (Figure 1A,B), and DEGs were intersected with MT for analysis, yielding 35 candidate genes (Figure 1C).

### 2.2. Screening of Signature Genes

To screen for stroke signature genes, we applied the previously obtained 35 candidate genes to three commonly used feature selection algorithms: SVM-RFE, RF, and LASSO. Twelve signature genes were obtained by screening using LASSO regression analysis (Figure 2A,B). SVM identified 35 gene markers with an accuracy of 0.864 (Figure 2C,D). Subsequently, the 35 candidate genes were ranked by importance scores using the random forest method, and 28 characteristic genes were screened (Figure 2E). Finally, we performed an intersection analysis of the results of the three methods and obtained 12 common key feature genes (Figure 2F).

### 2.3. Construction of the PPI Network and XGBoost Model

The 12 screened genes were uploaded to the STRING database, and the protein–protein interaction (PPI) network among them was constructed (Figure 3A). After network analysis, we found that eight of the genes were at the core of the network with more connecting edges and might play key regulatory roles in stroke pathogenesis (Figure 3A). Therefore, we constructed an XGBoost machine learning model for stroke prediction. The dataset was first divided into 70% training set and 30% test set. The key hyperparameters of the XGBoost model, such as the learning rate and the maximum depth of the tree, were iteratively tuned using the caret software package (6.0-94). We found that the AUC of the model reached 0.806 (95% CI: 0.778–0.833) (Figure 3B). This suggests that the XGBoost model based on these eight central genes has a high stroke prediction accuracy.

### 2.4. Enrichment Analysis

GO enrichment analysis of 74 DEGs revealed 350 biological processes (BP), 2 cellular components (CC) and 25 molecular functions (MF), as shown in Appendix A. The top 10 GO items were listed (Figure 4A,B). According to KEGG analysis, DEGs were significantly enriched in IL-17 signaling pathway, TNF signaling pathway, fluid shear stress, and atherosclerosis (Figure 4C,D). In addition, using DO enrichment analysis, we identified a total of 399 disease entries associated with DEGs (Figure 4E,F). Among these, 4 DEGs were significantly enriched in the category of cerebrovascular disease (Figure 4G), suggesting that they may be hub genes for stroke.

### 2.5. Expression Levels and Diagnostic Significance of Key Genes

We evaluated the potential value of 4 key hub genes (*ADM*, *PTGS2*, *VCAN*, and *MMP9*) in stroke diagnosis. In the combined dataset, the following AUC values for these genes were obtained: *ADM*, 0.777 (95% CI: 0.674–0.871); *MMP9*, 0.749 (95% CI: 0.647–0.845); *PTGS2*, 0.703 (95% CI: 0.583–0.800); and *VCAN*, 0.724 (95% CI: 0.615–0.828) (Figure 5A–D). In the independent validation set, their AUCs were further improved as follows: *ADM*, 0.734 (95% CI: 0.603–0.853); *MMP9*, 0.820 (95% CI: 0.715–0.911); *PTGS2*, 0.877 (95% CI: 0.795–0.943); and *VCAN*, 0.941 (95% CI: 0.889–0.983) (Figure 5E–H). These results indicated that these hub genes had good diagnostic ability for stroke. Meanwhile, box plot analysis showed that the expression levels of these four hub genes were significantly higher in the stroke samples than in the normal control group (Figure 5I–L). This further confirmed their important regulatory roles in stroke pathogenesis.

### 2.6. miRNA–TF–mRNA Analysis

To understand the overall framework of gene regulation and the regulatory relationships, we predicted the potential regulator miRNAs and transcription factors of these hub genes using the NetworkAnalyst database. It was found that miR-144 might simultaneously regulate *ADM*, *PTGS2*, and *VCAN*, while the transcription factor FOS might interact with *ADM*, *PTGS2*, and *MMP9* (Figure 6). This provided a basis for further elucidation of the gene regulatory network.

### 2.7. Hub Gene Enrichment

Using single gene GSEA analysis, we found that these hub gene-enriched pathways were mainly involved in immune-related processes such as allograft rejection, autoimmune diseases, and graft-versus-host disease (Figure 7). This further supports their key role in stroke pathogenesis.

### 2.8. Correlation of Hub Genes with Immune Infiltrating Cells

We analyzed immune cell infiltration in the stroke (IS) group and normal control group using the CIBERSORT algorithm (Figure 8A,B). The results showed that the abundance of M0 macrophages, activated mast cells, and neutrophils was significantly increased, while the abundance of CD8+ T cells and resting dendritic cells was significantly decreased in the IS group compared to the normal group. This suggests that inflammatory immune cells play a key role in the pathogenesis of stroke. Further studies revealed that the expression levels of these four hub genes (*ADM*, *PTGS2*, *VCAN*, and *MMP9*) were significantly positively or negatively correlated with the abundance of most immune cells (Figure 8C–F). We suggest that these key genes may be involved in regulating the infiltration and function of immune cells, thereby influencing the development of stroke.

### 2.9. scRNA Profiling in IS

To investigate the distribution of the four key genes in different cell types in stroke samples, we classified and cellularly annotated the single-cell samples we obtained. We identified 21 clusters in the GSE174574 dataset with nine different cell types: oligodendrocytes, monocytes, microglia, macrophages, epithelial cells, endothelial cells, dendritic cells, cardiomyocytes, and astrocytes (Figure 9A,B). Notably, we observed the predominant distribution of hub genes in microglia, macrophages, and dendritic cells in GSE174574 stroke samples (Figure 9C,D).

Next, we analyzed intercellular communication between different types of immune cells in the stroke setting using CellChat (Figure 10A,B). Activation of the MK pathway induces the expression of antioxidant genes, such as NRF2 target genes, increasing the antioxidant capacity of cells [23]. Therefore, we examined the communication network of the MK pathway in immune cells in stroke samples (Figure 10C,D). By analyzing the altered communication network of the MK pathway, we can better understand the oxidative stress induced by stroke and the response process of the immune system. This may help to further elucidate the pathogenesis of stroke and provide strategies for the development of targeted antioxidant therapeutic strategies.

### 2.10. Drug Prediction for IS Therapy

In summary, this study successfully predicted potential therapeutic small molecule compounds by submitting differentially expressed genes to the Connectivity Map (CMAP) database (all connectivity scores > 0.7) (Figure 11), and we selected the two drugs with the largest positive and negative scores, TAK-715 and prostratin, respectively. TAK-715 is a potent p38 MAPK inhibitor, and prostratin is a natural diterpenoid, an activator of PKC.

### 2.11. Molecular Docking of the Four Core Genes

Molecular docking analysis showed that the four hub genes formed good binding conformations with both prostratin and TAK-715 (Figure 12), suggesting that they may be potential targets of these two compounds. The inverse correlation between the binding energy and the stability of the binding conformations provides a basis for evaluating the interactions between these small molecules and their targets. This provides an important theoretical basis for further drug development and mechanism of action studies.

### 2.12. Key Gene Validation

First, we successfully established a rat MACO (middle artery permanent occlusion) ischemic stroke model. TTC (2,3,5-triphenyltetrazolium chloride) staining revealed that the area of cerebral infarction in the MACO model group was significantly larger than that in the control group (Figure 13A). To further investigate the changes in the expression of key genes in this model, we collected peripheral blood from rats in the MACO and control groups and extracted RNA for qPCR analysis. The results showed that four genes, *ADM*, *PTGS2*, *MMP9*, and *VCAN*, were significantly upregulated in MACO compared with the control group (Figure 13B). The expression changes in these genes provided important clues for further exploration of the pathogenesis and therapeutic targets of ischemic stroke and laid the foundation for subsequent translational medicine research.

## 3. Discussion

As the third leading cause of death and the first cause of disability, stroke is a major public health problem worldwide [24]. Stroke represents an escalating public health challenge due to increased morbidity, mortality, and disability. Although endovascular reperfusion therapy improves the prognosis of some patients, the prognosis of IS remains poor due to the narrow therapeutic window, the potential risk of hemorrhage, and subsequent reperfusion injury [25]. Meanwhile, the pathogenesis of stroke is a complex process involving the interaction of multiple factors. The central nervous system as well as environmental, systemic, genetic, and vascular factors all play important roles. To better prevent and treat stroke, we need to fully understand its underlying mechanisms, search for novel diagnostic biomarkers and therapeutic targets for IS, and develop innovative early detection and treatment strategies. By studying this disease in depth, we can take more targeted preventive and interventional measures to reduce stroke morbidity and mortality.

Melatonin (N-acetyl-5-methoxytryptamine) is an evolutionarily conserved hormone that is mainly synthesized by the pineal gland. It is also found in other parts of the body, including the retina, skin, gastrointestinal tract, and bone marrow [26]. Its primary function is to regulate circadian rhythms in mammals, and it is implicated in a variety of physiological processes, including emotional behavior, blood pressure regulation, ovarian physiology, and osteoblast differentiation [27]. Melatonin’s ability to readily cross the blood–brain barrier (BBB) makes it an ideal neuroprotective agent. Numerous studies have linked melatonin to a variety of disease models, including stroke, traumatic brain injury, Alzheimer’s disease, and Parkinson’s disease [28]. After a stroke, especially an ischemic stroke, it leads to an interruption of blood flow to the brain, which in turn triggers a severe oxidative stress and inflammatory response [29]. In this situation, the overproduction of ROS and RNS not only damages cell membranes, proteins, and DNA, but also leads to mitochondrial dysfunction, which ultimately triggers neuronal apoptosis and brain tissue damage [30]. Numerous studies have shown that melatonin can scavenge various types of reactive oxygen species and free radicals by directly scavenging ROS/RNS, activating the antioxidant system, targeting mitochondria, inhibiting pro-oxidant enzymes [31], and activating antioxidant enzyme synthesis [32] to reduce protein or DNA damage.

Meanwhile, melatonin, as an important circadian regulator, not only plays a key role in sleep regulation, but may also be associated with the timing of ischemic stroke. It is noteworthy that most human ischemic strokes occur in the early morning hours [33]. This phenomenon contrasts interestingly with the circadian pattern of melatonin secretion: melatonin levels rise during the night and fall rapidly in the early morning. We hypothesize that this rapid change in melatonin levels may be related to the increased incidence of stroke in the early morning. The sharp drop in melatonin levels in the early morning may lead to changes in the expression of these protective genes, thereby increasing the risk of stroke. For example, decreased expression of ADM may attenuate vasodilatory and neuroprotective effects, while increased expression of MMP9 and PTGS2 may exacerbate inflammatory responses and blood–brain barrier damage. Such changes in gene expression patterns, in combination with other physiological changes (e.g., increased blood pressure, increased coagulation activity, etc.) that are specific to the early morning, may together explain the high prevalence of ischemic stroke in the early morning. In addition, physiological changes that accompany the sleep–wake transition, such as fluctuations in blood pressure and changes in clotting factor levels, may also increase the risk of stroke. Future studies should further investigate the precise relationship among melatonin levels, sleep cycles, and the timing of stroke, which may provide an important basis for the development of time-specific preventive and therapeutic strategies.

For decades, pathophysiological studies of IS have focused on apoptosis and neuroregeneration, with little clinical progress. With great advances in bioinformatics, microarray data can be used to reveal key genes, interaction networks, and pathways that explain IS. In this study, we analyzed two GEO datasets to compare stroke and control samples, followed by further use of PPI, enrichment analysis, and various machine learning algorithms to identify four core diagnostic biomarkers. GSEA was used to investigate the function of these core genes in IS. The CIBERSORT algorithm performed a comprehensive analysis of the infiltration levels of 22 immune cell types and found significant correlations between the expression levels of *ADM*, *MMP9*, *PTGS2*, and *VCAN* and the infiltration levels of several immune cells that may contribute to the development of IS by affecting the immune milieu. We also developed an XGBoost model to predict the risk of IS. Currently, the diagnosis and treatment of IS is mainly based on clinical trials with inadequate response rates. Accurate diagnosis of IS and early preventive measures are essential to alleviate suffering and improve disease prognosis. In the available IS dataset, the AUCs of the four pivotal genes were all greater than 0.7, suggesting that the pivotal genes have good diagnostic value. Meanwhile, the AUC value of the XGBoost model reached 0.806. The sensitivity and specificity of the model suggest that it has a good ability to both identify high risk individuals and exclude low risk individuals. Future studies should investigate model performance in these subgroups and consider developing tailored models for specific populations. Overall, these biomarkers identified based on histological analysis have higher diagnostic accuracy and application potential than traditional diagnostic methods based on clinical symptoms, which may contribute to the early detection and precise intervention of stroke.

To further validate these findings, we investigated the expression patterns of these genes in different cell types using single-cell transcriptome sequencing data. Hub genes were mainly distributed in microglia, macrophages, and dendritic cells in stroke samples. Previous studies have shown that MAPK contributes to the regulation of inflammatory responses, cytokines, apoptosis, and death in ischemic and hemorrhagic brain injury [34,35]. Our single-cell analyses further revealed the inter-regulation of the MAPK pathway among microglia, macrophages, and dendritic cells, which plays a key role in stroke pathogenesis. We also analyzed the cellular communication network among these cell types using cell chat and focused on the regulatory role of the MAPK pathway therein, providing an important basis for further exploration of new therapeutic targets for stroke.

To identify new potential drug candidates for the treatment of IS and to explore the molecular basis of IS pathogenesis, we used the CMAP database for prediction (connectivity score > 0.7). TAK-715 is a potent p38 MAPK inhibitor. While endogenous inhibition of p38 MAPK rescued hippocampal apoptosis, reduced ischemic penumbra, and ameliorated neurobehavioral deficits, it is neuroprotective against IS [36]. Prostratin is a natural diterpene that is an activator of PKC. The PKC-related protein AKT is heavily involved in the regulation of apoptosis through several downstream pathways [37]. Considerable evidence supports the neuroprotective role of Akt in cerebral ischemia, and Akt expression has been found to be associated with neuronal repair, reduced oxidative stress, and reduced neuronal apoptosis [38]. In addition, we performed molecular docking analysis using hub genes and drugs to investigate intermolecular interactions by assessing binding energies to small molecules in the protein–ligand network, which enabled the mining of potential therapeutic targets. The results showed that *ADM*, *MMP9*, *PTGS2*, and *VCAN* have good binding affinity to TAK-715 and prostratin, suggesting that they are potential targets of TAK-715 and prostratin. However, further studies are needed to elucidate the specific regulatory mechanisms and provide an important theoretical basis for the development of new IS therapeutics. Finally, four key genes, *ADM*, *MMP9*, *PTGS2*, and *VCAN*, were significantly upregulated in IS tissues as verified by qPCR experiments, which was consistent with the results of the preliminary bioinformatics analysis. The important role of these genes in the pathogenesis of IS was further confirmed.

Using a bioinformatics approach, four hub genes (*ADM*, *MMP9*, *PTGS2*, and *VCAN*) closely associated with IS were identified for the first time. *ADM* is a prohormone that is cleaved to form two bioactive peptides with multiple functions, including vasodilation, regulation of hormone secretion, promotion of angiogenesis, and antimicrobial activity [39]. It regulates vascular integrity by acting through various receptors. It helps maintain vascular structure and modulates angiogenesis as a vasoprotective agent against vascular injury, and these effects have been observed in both acute and chronic cerebral ischemia. The extracellular matrix (ECM) plays an important role in tissue repair, including cardiac regeneration [40], and as *VCAN* is a major component of the ECM, studies have shown that increasing *VCAN* leads to cardiac regeneration and recovery of cardiac function [41]. This has important implications for the treatment of stroke. Prostaglandin endoperoxide synthase (*PTGS2*), also known as *COX-2*, is an important response gene that has been implicated in maintaining homeostasis and regulating inflammation in normal tissues in vivo [42], and the association of polymorphisms in the *COX-2* gene with ischemic stroke has been explained in several studies [43]. Silencing the expression of *PTGS2* plays an important role in angiogenesis and inhibits apoptosis [44], so *PTGS2* can be used as a therapeutic target for ischemic stroke. *MMP9* is a gelatinase that can degrade the extracellular matrix and the basement membrane of the blood–brain barrier, increasing its permeability and leading to brain edema and inflammatory cell infiltration [45]. Taken together, these four key genes play critical roles in the pathogenesis of stroke, including regulation of vascular integrity, extracellular matrix remodeling, and regulation of the inflammatory response, providing novel biomarkers and potential targets for the diagnosis and treatment of stroke. This is of great translational value in improving the prognosis and quality of life of stroke patients.

In the present study, we investigated the expression changes of melatonin-related genes during ischemic stroke (IS) using the rat middle artery permanent occlusion (MACO) model and identified four key genes. However, due to interspecies differences in gene regulatory mechanisms, these findings need to be further validated for relevance in specific patient groups or stroke subtypes before clinical application. Next steps should include further ex vivo investigation of the specific relationship between melatonin and these key genes. In addition, the choice of a young animal model in this study may limit the direct applicability of the findings to the elderly IS patient population, a factor that needs to be considered in future studies. In conclusion, although this study has identified some potential therapeutic targets, further clinical validation and in-depth mechanistic studies are required before clinical translational application.

## 4. Materials and Methods

### 4.1. Data Acquisition and Pre-Processing

In this study, we aimed to investigate the role of melatonin-related genes (MT) in the pathogenesis of IS by analyzing peripheral blood samples from stroke (IS) and normal controls. We obtained three microarray datasets (GSE22255, GSE16561, and GSE58294) and one single-cell RNA sequencing (scRNA-seq) dataset (GSE174574) from the GEO database. Meanwhile, we obtained the melatonin-related (MT) gene set from the Coremine Medical database. Among them, GSE22255 and GSE16561 were used as screening datasets, and GSE58294 and GSE174574 were used as validation datasets. After the steps of data pre-processing, batch effect correction, and expression value aggregation, we obtained the final gene expression matrix. Details are shown in Appendix A, and the overall study flow is shown in Figure 14.

### 4.2. Identification of Differentially Expressed Genes

To identify differentially expressed genes (DEGs) between healthy and IS samples, we used the limma package in R for differential expression analysis. Screening criteria were set as *p*-value < 0.05 and |log2(fold change)| > 0.5. Heatmaps and volcano plots were visualized using the ggplot2 package to further illustrate the expression characteristics of DEGs. Next, we used the Venn diagram package to analyze the intersection of the screened DEGs with MT to identify melatonin-related genes that were differentially expressed in the pathogenesis of IS in order to reveal the potential regulatory role of melatonin in the pathogenesis of IS.

### 4.3. Support Vector Machine, Random Forest, and Least Absolute Shrinkage with Selection Operator Model Construction

In order to improve the diagnostic predictive ability of IS and to discover potential biomarkers, we applied three commonly used machine learning models to screen and classify the feature genes for analysis using the SVM-RFE method with the e1071 package (1.7-16) [46], the RF algorithm with the randomForest R package (4.7-1.2) [47], and the LASSO algorithm with the glmnet package (4.1-8) [48] for feature gene classification. Finally, we performed a Venn analysis of the feature genes screened using the above three machine learning models to find the candidate biomarker genes jointly identified by the three methods.

### 4.4. PPI Analysis and XGBoot Model Construction

To investigate the role of candidate genes in the pathogenesis of stroke (IS) in depth, the candidate genes were uploaded to the STRING database, and the PPI network diagram was constructed to identify the potential key genes. Meanwhile, the XGBoost algorithm was applied to model the classification prediction of the candidate genes, and the pROC package (1.18.5) [49] was used to calculate the area under the ROC curve (AUC) value to assess the diagnostic accuracy of the model. An AUC of 0.5–0.7 was considered as low accuracy, 0.7–0.9 as moderate accuracy, and >0.9 as high accuracy.

### 4.5. Enrichment Analysis and Assessment of Hub Gene Diagnosticity

To determine the biological function and significance of the candidate genes, we performed GO, KEGG, and DO analyses using clusterProfiler (4.12.6) [50], org.Hs.eg.db (3.19.1), and DOSE packages (3.30.5). A *p*-value of <0.05 was used as a significance screening criterion to further explore the potential key genes. The diagnostic value of the key genes in IS was further evaluated, and we verified their expression in the screening and validation sets, plotted the expression box line graphs using the ggplot2 package (3.5.1), and calculated the ROC curves and AUC values using the pROC package.

### 4.6. miRNA–TF–mRNA Regulatory Network Analysis

The NetworkAnalyst online tool (https://www.networkanalyst.ca/, accessed on 15 March 2024) [51] was used to construct the interaction network diagrams between candidate genes and miRNAs and TFs to comprehensively elucidate the multi-level regulatory relationships of gene expression and also to predict new potential regulatory axes. Finally, we used Cytoscape software (3.8.0) to visualize the regulatory network.

### 4.7. Hub Gene Enrichment Analysis and Immune Infiltration

GSEA was used to determine the concordance of highly enriched gene sets. Enrichment analysis was performed to analyze the relevance of hub genes to disease states and their regulatory mechanisms [52]. In contrast, CIBERSORT is an inverse convolution algorithm that transforms the normalized gene expression matrix into the composition of infiltrating immune cells. We therefore compared the infiltration of 22 immune cells in normal and IS samples. LM22 was used as the reference expression signature, and 1000 permutations were performed to more accurately predict the immune cell composition. CIBERSORT output was defined as *p* < 0.05, and samples meeting the constraints were then selected for further analysis. The 22 infiltrating immune cells included B cells (naive B cells and memory B cells), T cells (CD8 T cells, naive CD4 T cells, memory resting CD4 T cells, memory activated CD4 T cells, follicular helper T cells, regulatory T cells and γδ T cells), NK cells (resting NK cells and activated NK cells), monocytes, macrophages (M0 macrophages, M1 macrophages, and M2 macrophages), dendritic cells (resting dendritic cells and activated dendritic cells), mast cells (resting mast cells and activated mast cells), eosinophils, and neutrophils. All 22 immune cell type fractions assessed were summed to 1 per sample [53].

### 4.8. Analysis of Single-Cell Data and Intercellular Communication

We obtained scRNA-seq data GSE174574 from the GEO database. In subsequent analyses, we used Seurat (v.5.0.1) [54] for quality control and downstream analysis of GSE174574. During quality control, we removed genes with fewer than 200 or more than 3000 copies, as well as genes expressed in fewer than 3 cells. Unique molecular identifier (UMI) counts of mitochondrial origin that exceeded 15% were then filtered out. Next, the data set was log-normalized, and the top 2000 highly variable genes (HVGs) were selected for typical correlation analysis. We then normalized the data using the ScaleData function and performed principal component analysis (PCA). Units were clustered using the first 10 principal components with a resolution parameter of 0.5. These principal components were also used to generate TSNE and UMAP plots [55]. Finally, we annotated the data using SingleR (2.4.1) and CellMarker 2.0 [56], followed by relevant visualization analyses. To explore in depth the signaling and regulatory mechanisms between different cell types, we used the CellChat package (v1.6.1) [57] for analysis, which can help us to better understand the dynamics of the cellular microenvironment during IS pathogenesis.

### 4.9. CMAP Analysis

CMAP analysis, as a high-throughput drug screening tool, contains 6100 instances of 1309 small molecule drugs, each containing the gene expression profile of a specific drug and its corresponding treatment, which can effectively link the gene expression data with the drug response and provide new directions for disease treatment, thus advancing research progress in related fields. Therefore, we carried out drug prediction. In this study, we used gene expression profiles to predict potential molecular compounds for IS therapy. First, up- and downregulated DEGs were uploaded to the CMAP database for analysis, and the system scored these genes according to their expression patterns, with scores ranging from −100 to 100. By comparing the scores, negative values indicated that the expression profile of the compound was negatively correlated with the disease state, predicting that it might have therapeutic potential. This analysis helps us screen out candidate compounds that merit further validation.

### 4.10. Protein–Ligand Interaction Analysis

To further validate the candidate compounds predicted using CMAP analysis, we performed protein–ligand molecular docking experiments. First, we obtained the 3D structures of the core target proteins from the Uniprot database, usually in PDB format. We ensured that the selected structure has a high resolution to improve docking accuracy and downloaded the SDF structure files of the CMAP-predicted small molecule compounds from PubChem and converted them to mol2 format using OpenBabel software (2.4.0). The mol2 format is suitable for AutoDock for docking analyses because it contains key information such as atom types, bond information, and partial charges. Molecular docking analysis was then performed on the protein receptor and the small molecule ligand using AutoDock software (v4.2.6), and the binding free energy of the two was calculated. The lower the binding energy value, the stronger the binding affinity. We used PyMol software (2.1) to visualize the docking results in order to visually analyze the interaction patterns of the compounds with the target proteins.

### 4.11. Animals and MACO Modelling

Wistar rats (male, 200–240 g) were provided by Beijing Viton Lever Laboratory Animal Co. and were housed under specific conditions with free access to food and water. The rats were randomly divided into two groups (*n* = 6 in each group): the MCAO and control groups. The MCAO model was established according to a previous report [58]. Briefly, rats were anaesthetized with 2% isoflurane. After a midline incision, the right common carotid and external carotid arteries were isolated, and the internal carotid artery was clamped. MCAO was then performed for 24 h. Peripheral blood was then collected from rats in the MCAO and control groups, and RNA was extracted for quantitative qPCR analysis to monitor changes in the expression of key genes. The effect of establishing the MCAO model was also evaluated.

### 4.12. RNA Extraction and qPCR

Total RNA was extracted from rat peripheral blood using TRIzol, and the first-strand cDNA of RNA was synthesized using first-strand Synthesis Master Mix (LABLEAD, Beijing, China). The primers used for cDNA amplification are listed in Table 1. The cDNA was mixed with SYBR Premix Ex Taq2 (TaKaRa, Kusatsu, Japan) and synthetic primers and subjected to real-time quantitative PCR. PCR conditions were selected according to the manufacturer’s protocol as follows: 2 min at 50 °C; 10 min at 95 °C; and 45 cycles of 10 s at 95 °C, 10 s at 60 °C and 15 s at 72 °C. PCR was performed in real time using the PrimeScript RT kit (TaKaRa, Japan). Relative mRNA expression levels were quantified by normalization to the expression of the internal reference GAPDH. Gene expression levels are expressed as fold change over control.

### 4.13. Statistical Analyses

R software (version 4.4.2) was used for data examination. All in vitro experiments were performed three times. Results were expressed as mean ± SD of three replicates. Differences between groups were compared using one-way analysis of variance (ANOVA) followed by Tukey’s post hoc test using GraphPad Prism 8 (GraphPad, La Jolla, CA, USA). Differences were significant when *p* < 0.05.

## 5. Conclusions

In this study, a systems bioinformatics approach was used to successfully identify four key melatonin genes in IS patients: *ADM*, *MMP9*, *VCAN*, and *PTGS2*. qPCR experiments verified that the expression profiles of these key genes showed significant aberrant regulation in IS patients, indicating that they play important roles in the pathophysiological mechanisms of IS. This finding provides a better understanding of the molecular mechanism of the role of melatonin in IS and lays the foundation for the development of novel therapeutic strategies and individualized medical treatment protocols with important clinical translational value.

## Figures and Tables

**Figure 1 ijms-25-11620-f001:**
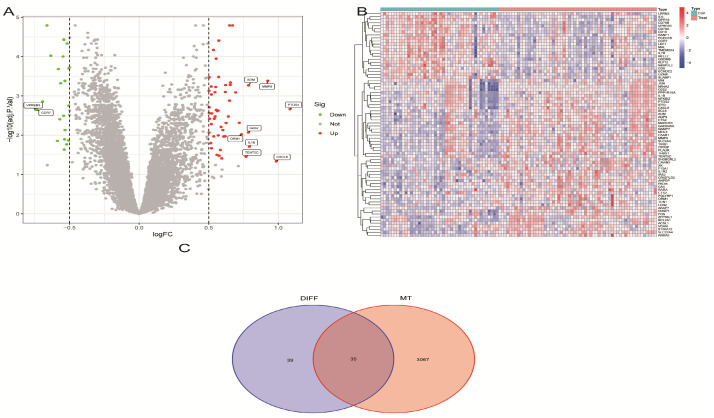
Screening of DEGs between IS and controls. (**A**) Venn diagram visualizing DEGs in IS and normal samples. (**B**) Heat map of DEGs. (**C**) Venn diagram of overlapping genes.

**Figure 2 ijms-25-11620-f002:**
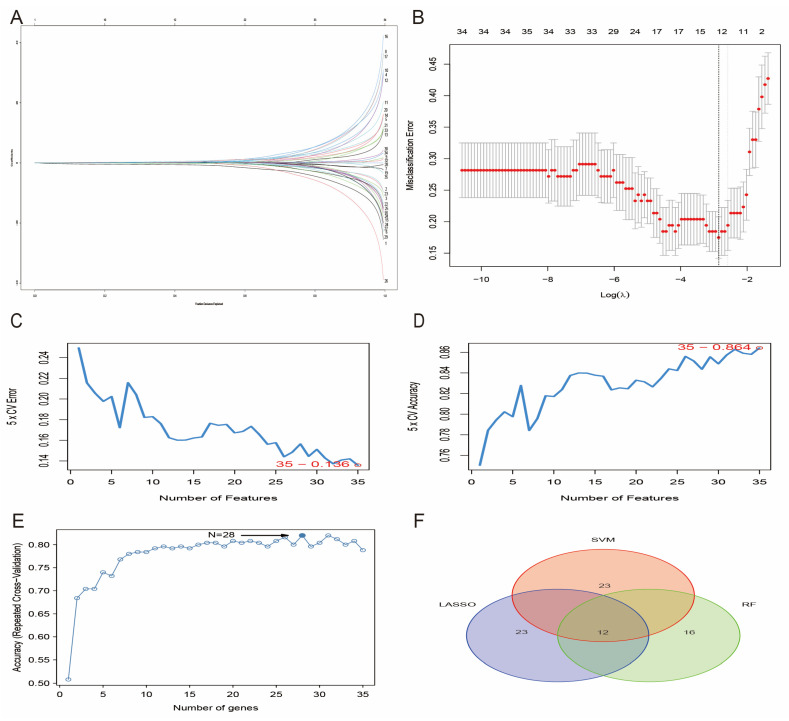
Identification of characterized genes. (**A**) LASSO lineage map of candidate genes. (**B**) Cross-validation in LASSO regression analysis to select the best tuning parameter log. (**C**) SVM-RFE analysis identified 35 feature genes with an error of 0.136. (**D**) Accuracy of 0.864. (**E**) In total, 28 feature genes were screened using a RF algorithm. (**F**) Venn diagram of 12 feature genes shared by SVM-RFE (pink), RF (green) and LASSO algorithms (blue).

**Figure 3 ijms-25-11620-f003:**
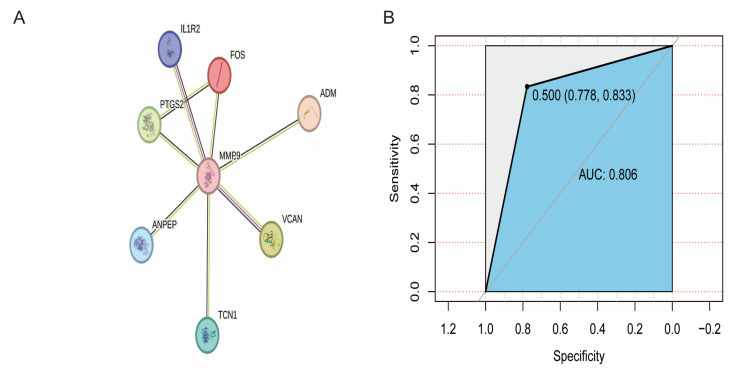
Feature genes. (**A**) Feature gene PPI network diagram. (**B**) Area under the blue ROC curve to assess XGBoost prediction accuracy.

**Figure 4 ijms-25-11620-f004:**
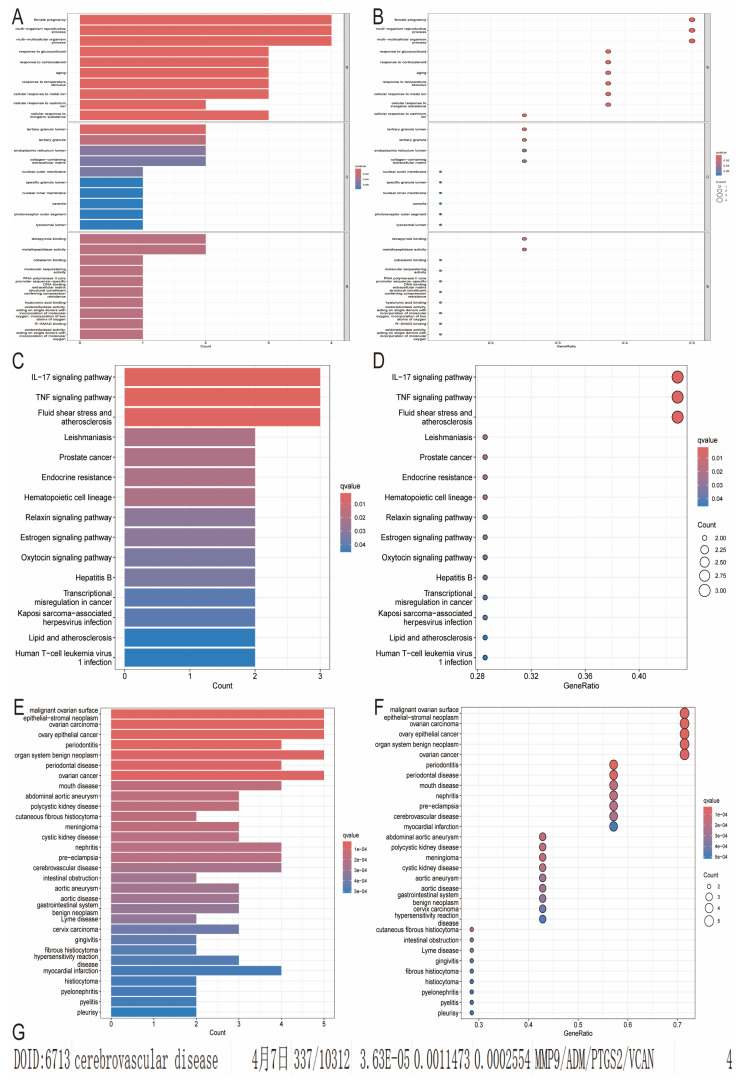
Enrichment of functional DEGs. (**A**,**B**) GO analysis. (**C**,**D**) KEGG pathway analysis. (**E**,**F**) DO analysis. (**G**) Enrichment of genes. (Non-English terms represent dates of analysed; 7 April).

**Figure 5 ijms-25-11620-f005:**
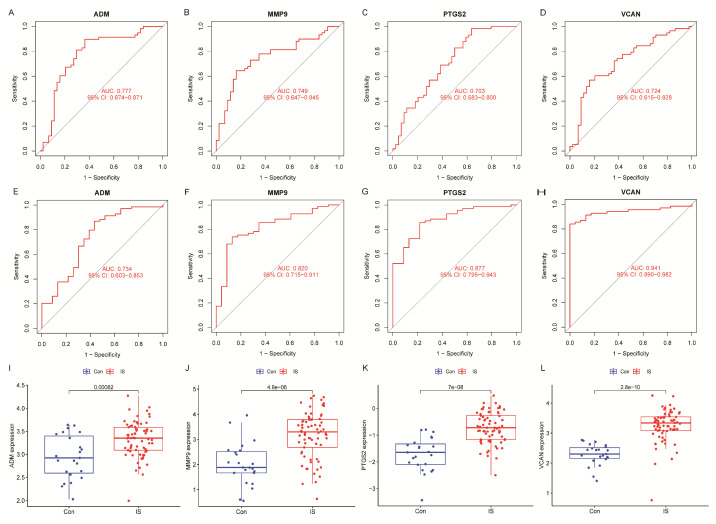
Expression analysis of key genes. (**A**–**D**) Screening the AUC of centrally diagnosed IS. (**E**–**H**) Validation of the AUC of centrally diagnosed IS. (**I**–**L**) Expression levels of four key genes in IS versus control.

**Figure 6 ijms-25-11620-f006:**
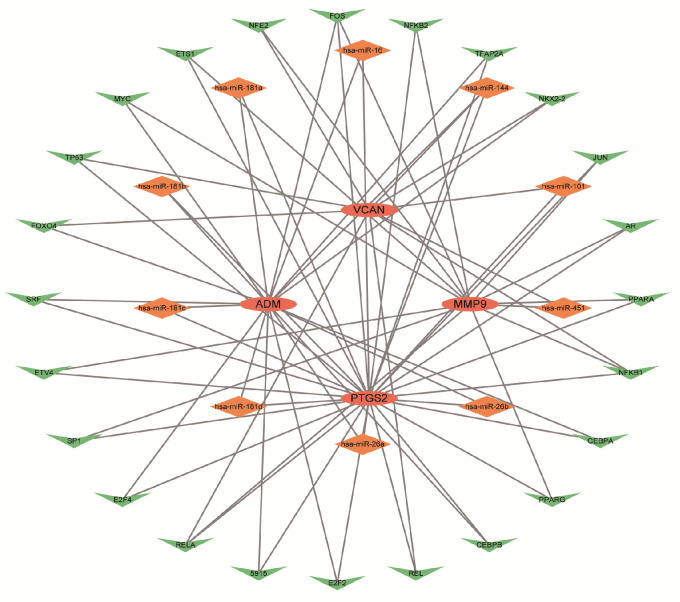
mRNA–miRNA–TF network diagram. Oval circles represent hub genes. Diamond squares represent miRNAs. Triangles represent TFs. Abbreviations: miRNA, microRNA; TF, transcription factor.

**Figure 7 ijms-25-11620-f007:**
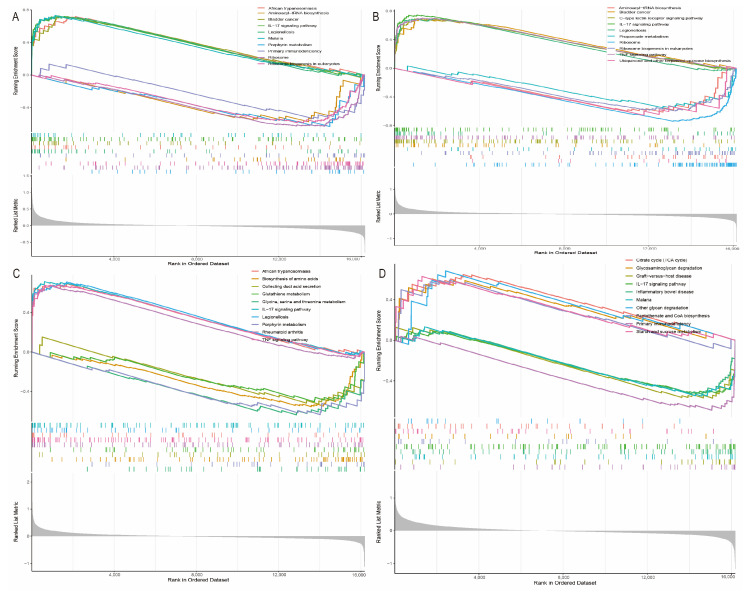
GSEA analysis of hub genes. Top 5 GSEA enrichments in the set of high and low expressed genes for (**A**) *ADM*, (**B**) *PTGS2*, (**C**) *VCAN*, and (**D**) *MMP9*.

**Figure 8 ijms-25-11620-f008:**
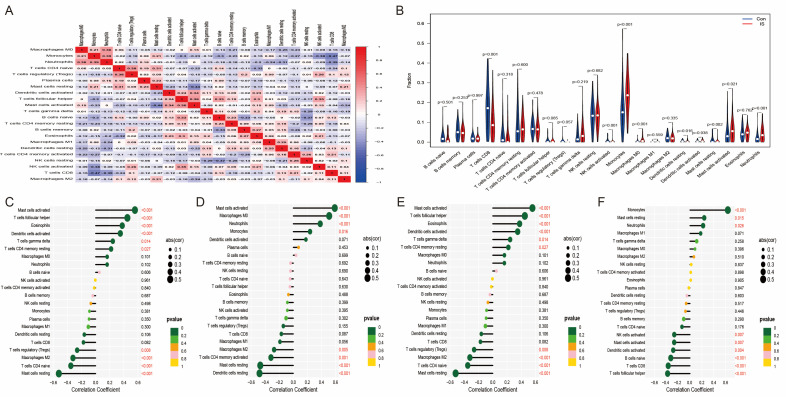
Distribution of immune cells in IS. (**A**) Correlation analysis of hub genes with immune cells. (**B**) Differences in infiltrating immune cells between IS and controls. (**C**) *ADM*, (**D**) *PTGS2*, (**E**) *VCAN*, and (**F**) *MMP9* correlation with immune cells.

**Figure 9 ijms-25-11620-f009:**
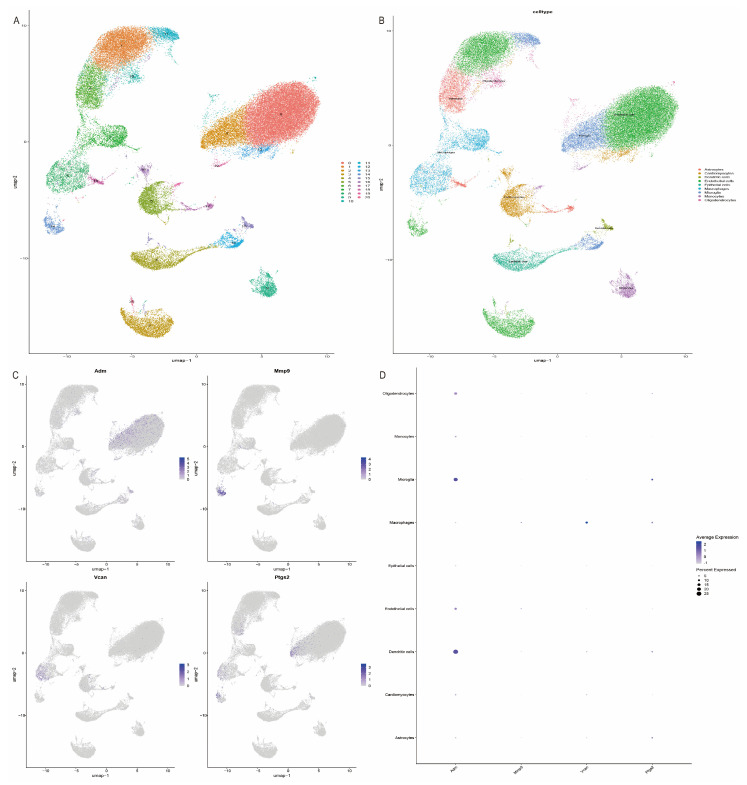
Validation of key genes in the single-cell dataset GSE174574. (**A**,**B**) UMAP plots showing clustering results for the GSE174574 dataset and the major cell types involved in stroke. (**C**,**D**) Distribution of genes across the various cell types in the GSE174574 dataset.

**Figure 10 ijms-25-11620-f010:**
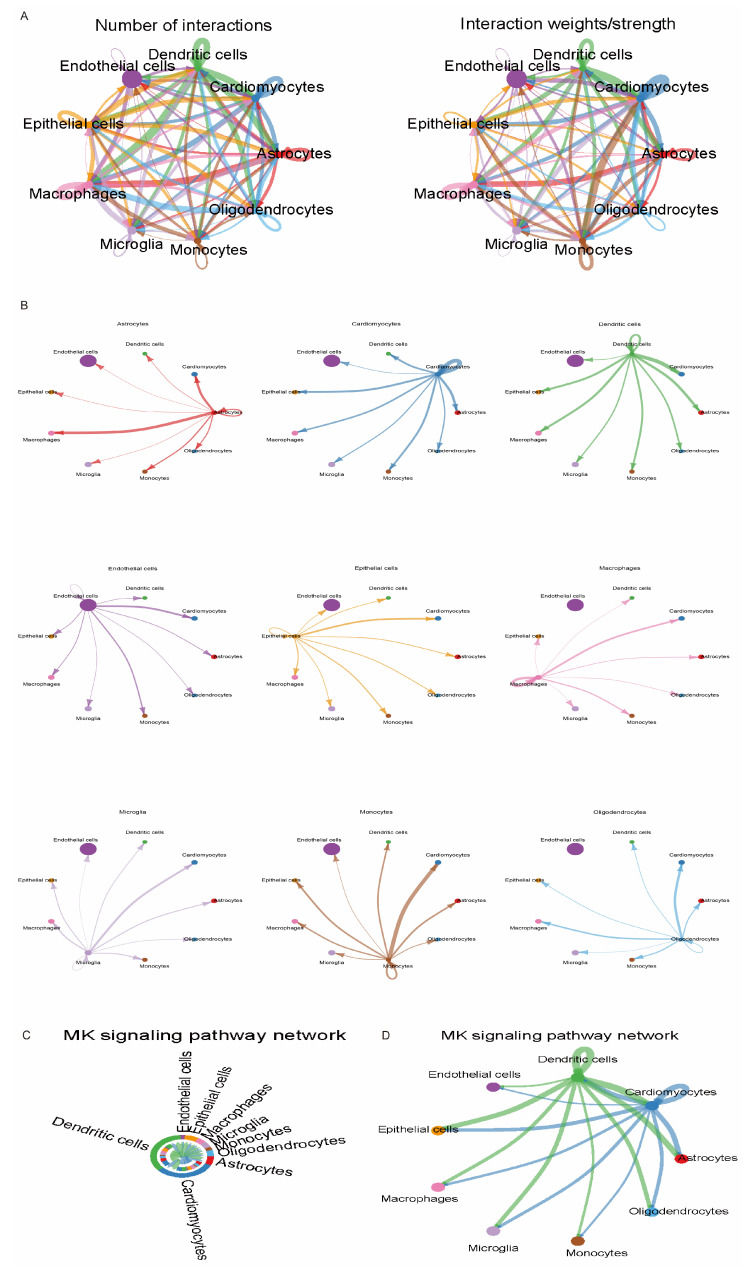
Intercellular communication of immune cells in the GSE174574 dataset. (**A**,**B**) Cross-talk analysis among GSE174574 stroke immune cells. (**C**,**D**) MK pathway network of the GSE174574 stroke immune cell population.

**Figure 11 ijms-25-11620-f011:**
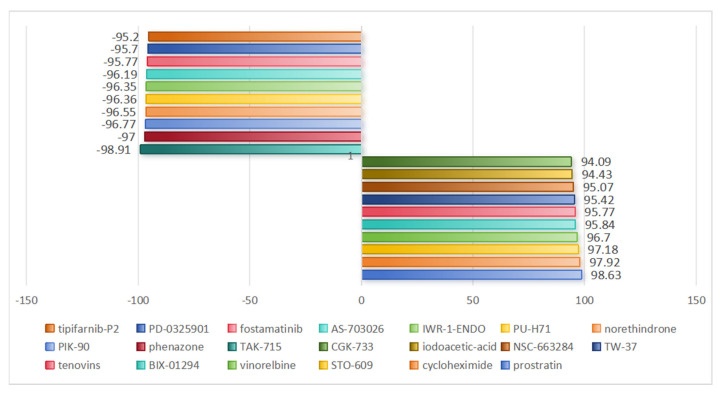
Drug prediction. CMAP instances by compound and cell line organization showing the most significant positive and negative correlation scores for IS effects. Correlation scores are shown on both sides, indicating the strength of the association.

**Figure 12 ijms-25-11620-f012:**
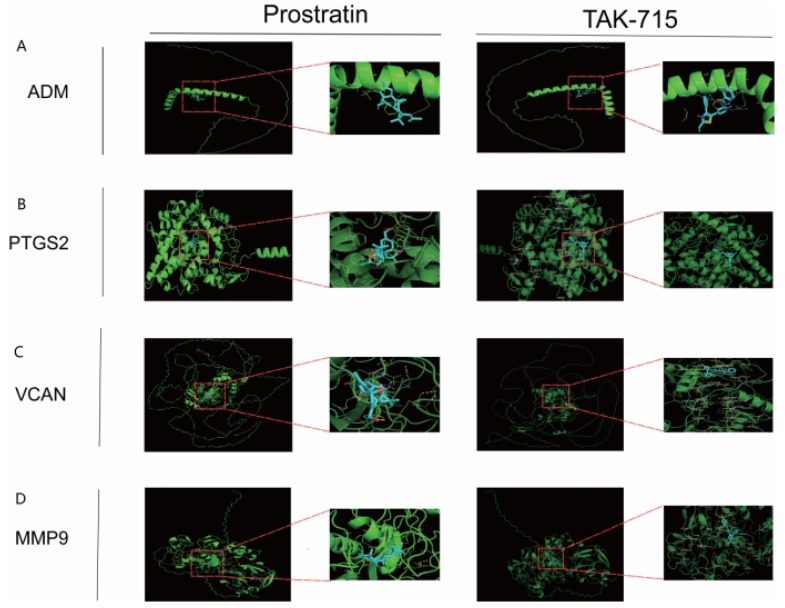
Molecular docking patterns of prostratin and TAK-715 with four target proteins. Binding effects of (**A**) ADM, (**B**) PTGS2, (**C**) VCAN, and (**D**) MMP9 with prostratin and TAK-715.

**Figure 13 ijms-25-11620-f013:**
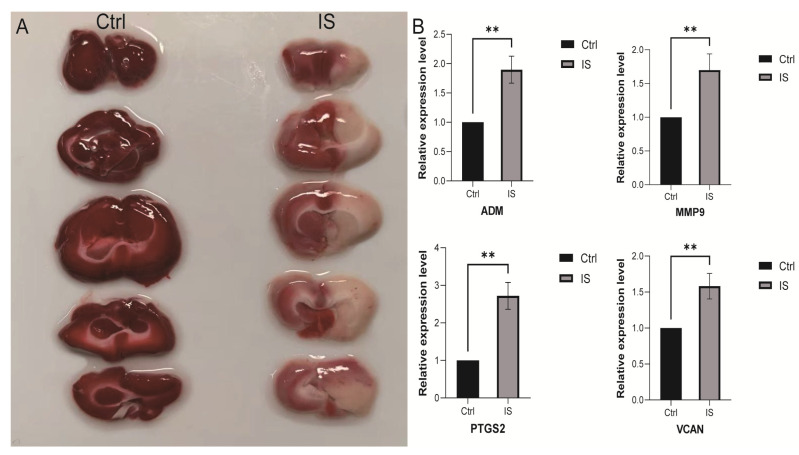
qPCR analysis. (**A**) TTC staining of rat brain injury, (**B**) gene expression levels of *ADM*, *MMP9*, *PTGS2*, *VCAN* in IS and ctrl. ** *p* < 0.01 versus Ctrl.

**Figure 14 ijms-25-11620-f014:**
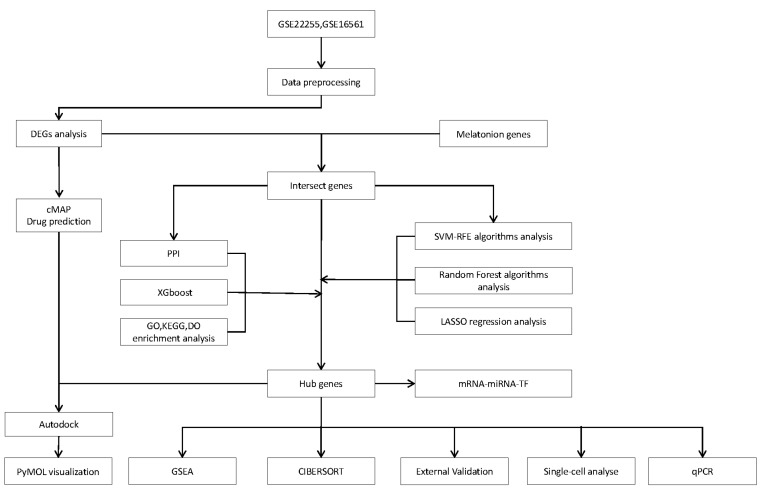
Study workflow.

**Table 1 ijms-25-11620-t001:** RT-PCR primers.

Genes		Primers (5′–3′)
Adm	F	TTGGACTTTGCGGGTTTTGC
	R	GATGCTCCGATACCCTGCTG
Mmp9	F	AGGGCCCCTTTCTTATTGCC
	R	CACATTTTGCGCCCAGAGAA
Ptgs2	F	ACGTGTTGACGTCCAGATCA
	R	GGCCCTGGTGTAGTAGGAGA
Vcan	F	GATGCCTACTGCTTTAAACCTAAAC
	R	AGCTCTCTCGGGTACCATGT

## Data Availability

Data are contained within the article and Appendix A.

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
