# Peer review of "Transcriptome Sequencing-Based Screening of Key Melatonin-Related Genes in Ischemic Stroke"

_ijms, 2024, doi:10.3390/ijms252111620_

Round 1
Reviewer 1 Report
Comments and Suggestions for Authors
An interesting work. I have some minor comments:
1) It should be useful to add a brief comment in the introduction on the relationship between ADM, VCAN, MMP9, and PTGS2 genes and melatonin.
2) Figure 4. Subheading for fig 4F is missing
3) Figure 13. There are missing letters or they did not correspond with the subfigures
Reviewer 2 Report
Comments and Suggestions for Authors
This manuscript presents a comprehensive and well-conducted study that uses transcriptome sequencing and bioinformatics tools to identify key melatonin-regulated genes in the context of ischemic stroke (IS). The study addresses an important topic with significant translational potential, particularly in identifying new diagnostic biomarkers and therapeutic targets for IS. Below is a detailed evaluation of the manuscript:
Strengths:
-
Novel Approach and Focus: The study combines transcriptome sequencing, bioinformatics, and machine learning to identify candidate genes related to melatonin and ischemic stroke. The inclusion of melatonin, a hormone with documented antioxidant and anti-inflammatory properties, in the context of IS is particularly relevant, as it opens up new therapeutic avenues.
-
Comprehensive Use of Bioinformatics Tools: The authors employed a variety of robust computational tools and algorithms such as SVM-RFE, Random Forest, and LASSO, which added rigor to the identification of candidate genes. Moreover, the integration of PPI networks, XGBoost model, and single-cell RNA sequencing data adds depth to the analysis.
-
Validation of Results: The validation of candidate genes (ADM, PTGS2, MMP9, and VCAN) through qPCR in an animal model strengthens the credibility of the findings. Additionally, the study's use of ROC curve analysis to assess the diagnostic accuracy of these genes further demonstrates the reliability of the results.
-
Enrichment and Pathway Analyses: The GO, KEGG, and DO enrichment analyses, as well as GSEA and immune infiltration analyses, provided a functional understanding of the role of these genes in stroke. This multi-dimensional approach strengthens the overall conclusions regarding the molecular mechanisms underlying IS.
-
Therapeutic Implications: The molecular docking and drug prediction analyses using the CMap database are valuable additions, as they propose potential therapeutic compounds (TAK-715 and Prostratin) that could be explored for IS treatment. The molecular docking analysis lends further support to the potential for developing these compounds as therapeutic agents.
Points for Improvement:
-
Data Presentation: While the figures, such as the volcano and heat maps, are well-crafted, some of the figures (especially the Venn diagrams and PPI network figures) are rather dense and may not be immediately clear to all readers. Additional clarification, labels, or a more user-friendly presentation of these figures might improve accessibility and comprehension.
-
Mechanistic Insight: Although the study identifies key genes and their roles in IS, more detailed mechanistic insights regarding how these genes specifically interact with melatonin in IS pathophysiology are needed. While the authors mention the role of melatonin in antioxidant and anti-inflammatory responses, direct experimental evidence linking melatonin to the modulation of the identified genes (ADM, PTGS2, MMP9, VCAN) is limited. Further in vitro or in vivo studies demonstrating this relationship would enhance the overall impact of the study.
-
Animal Model Limitations: The study focuses on gene expression in a rat MACO (middle artery permanent occlusion) model of IS, which is a well-established model. However, the authors should discuss the limitations of translating findings from animal models to human clinical settings. Differences in gene regulation and stroke pathophysiology between species may limit the direct applicability of these findings to humans.
-
Clinical Relevance: The authors highlight the potential use of the identified genes as diagnostic biomarkers. However, it would be beneficial to elaborate on how these biomarkers could be implemented in clinical practice. For instance, are there specific patient populations or stroke subtypes where these biomarkers could be most useful? Additionally, some discussion of the timeline for biomarker detection post-stroke and its clinical implications would strengthen the discussion on translation to clinical use.
-
Broader Context of Melatonin Research: The authors briefly mention previous studies linking melatonin to neuroprotection in stroke models. A more comprehensive review of the literature on melatonin’s role in neurological diseases, particularly stroke, would help situate this study within the broader field of melatonin research. This could also involve a more in-depth discussion of the therapeutic potential of melatonin and its clinical trials in stroke or other neurological conditions.
-
Discussion of XGBoost Model: While the XGBoost machine learning model is a useful tool for stroke prediction, the discussion lacks details on the specific performance of the model across different datasets. The authors mention an AUC of 0.806, but further elaboration on model sensitivity, specificity, and potential overfitting (if applicable) would be valuable for evaluating the robustness of the model in clinical settings.
Minor Comments:
- In the introduction, the authors might consider briefly explaining the complexity of IS at a molecular level, possibly including how ischemia leads to oxidative stress, inflammation, and apoptosis.
- Some sections, such as the drug prediction and molecular docking, could benefit from more detailed methodology to allow replication of the analysis.
- The title could be more specific by mentioning the identified genes (ADM, PTGS2, MMP9, VCAN) or the use of bioinformatics tools to attract more targeted readership.
- Some grammatical errors, such as in line 7 ("isczntified"), should be corrected for clarity.
Conclusion:
This manuscript presents an in-depth bioinformatics analysis to identify melatonin-related genes involved in ischemic stroke. The research is scientifically sound, and the findings are promising, particularly regarding the identification of new diagnostic biomarkers and therapeutic targets. The integration of transcriptome sequencing, machine learning models, and experimental validation provides a solid foundation for future studies. Addressing the points raised above could enhance the clarity and impact of the work.
Recommendation: Accept with minor revisions.
Reviewer 3 Report
Comments and Suggestions for Authors
Comments:
- The authors should mention the full names of the genes and their main functions in the abstract. For example, MMP has no relevance to melatonin.
- The authors should note that most ischemic strokes in humans occur during the early hours of the morning (see doi: 10.1016/j.expneurol.2009.12.023). Since melatonin is known as the sleep hormone, the authors should focus the discussion on this aspect.
- The authors have used young instead of old animals, a serious weakness of the study
English is OK
Reviewer 4 Report
Comments and Suggestions for Authors
The research topic is current and very interesting. It concerns an important health problem for many populations: stroke.
The research methodology is advanced and includes the tools of genomics, transcriptomics and microarray analyses, molecular modeling, advanced data analysis techniques and machine learning.
Using these modern tools, the authors confirmed previous observations regarding the role of immunological, inflammatory and oxidative stress-related processes in the pathogenesis and clinical course of stroke and indicated cells that play a dominant role in the pathogenesis of stroke.
In addition, they identified diagnostic biomarkers characterized by quite high sensitivity and specificity (the AUC value of the XGBoost model reached 0.806). These biomarkers identified based on 291 histological analysis have higher diagnostic accuracy and application potential than traditional diagnostic methods based on clinical symptoms, which may contribute to early detection and precise intervention of stroke.
They showed a correlation between the expression of several key melatonin-regulated genes associated with IS (ADM, MMP9, PTGS2 and VCAN) and the intensity of immune cell infiltration, which may contribute to the development of stroke. Four hub genes (ADM, MMP9, PTGS2 and VCAN) closely associated with IS were identified for the first time. These genes can play critical roles in the pathogenesis of stroke. They encode proteins that are involved in vasodilatation, angiogenesis and vasoprotection. VCAN is a component of extracellular matrix that is involved in tissue repair. PTGS2 (COX-2) is implicated in regulation of inflammation, and MMP9 can degrade the ECM and increase permeability of blood-brain barrier, leading to brain oedema and infiltration of inflammatory cells.
The research results presented in this manuscript may have great translational value in improving the diagnostics, prognosis and quality of life of stroke patients.
The research results will have to be confirmed in clinical trials - the results of experimental studies involving animals (mice) are not always reflected in clinical conditions. It would be worth mentioning this in more detail in the discussion.
The manuscript is well written, with a proper introduction, description of the methodology, analysis of the results. The discussion is interesting and closely relates to the obtained results.
Round 2
Reviewer 3 Report
Comments and Suggestions for Authors
The authors have successfully addressed my observations. The manuscript can be published in its present form
Comments on the Quality of English LanguageEnglish is good for publication purposes